# Preparation of Selenium-Enriched Yeast by Re-Using Discarded *Saccharomyces cerevisiae* from the Beer Industry for Se-Supplemented Fodder Applications

**Guojie Wu [1,\*], Fei Liu [2], Xiaowen Sun [3], Xuegui Lin [1], Feng Zhan [4] and Zhihuan Fu [1]**

[1]  School of Chemistry and Chemical Engineering, Zhongkai University of Agricultural Engineering, Guangzhou 510225, China

[2]  School of Science, Rensselaer Polytechnic Institute, Troy, MI 12180, USA

[3]  State Key Laboratory of Agricultural Microbiology, College of Life Science and Technology, Huazhong Agricultural University, Wuhan 430070, China

[4]  School of Chemical Engineering and Light Industry, Guangdong University of Technology, Guangzhou 510006, China

\*  Correspondence: gjwu@gdut.edu.cn; Tel.: +86-020-3417-2870

**Abstract:** Both inorganic and organic selenium (Se) can prevent and treat various diseases caused by Se deficiency. However, organic Se has less toxicity and a higher absorption rate than inorganic Se. In this study, inorganic Se ($Na_2SeO_3$) was bio-transformed into Se-enriched discarded beer yeast (Se-enriched DB-yeast) through fermentation accumulation by re-using discarded *Saccharomyces cerevisiae* from the beer industry for Se-enriched fodder application. Through a single-factor experiment and $L_9(3^4)$-orthogonal test for optimization of fermentation conditions, the Se content and biomass of Se-enriched DB-yeast were calculated as 14.95 mg/L and 7.3 g/L, respectively, under the optimized condition. The total amino-acid content of Se-enriched DB-yeast was increased by 9.9% compared with that from DB yeast. Additionally, alkaline amino-acid content was increased, whereas acidic amino-acid and sulfur-containing amino-acid contents were decreased. Reducing capacity, hydroxyl radical removal capacity, and sulfhydryl content after treatment with $H_2O_2$ of the Se-enriched DB-yeast extracted protein were obviously increased compared with those of the DB-yeast extracted protein. Mouse and genetically improved farmed tilapia (*Oreochromis niloticus*) (GIFT) bioassays showed that the Se sedimentation of organs and serum indexes after feeding Se-enriched DB-yeast-containing fodder were higher than those of DB-yeast-containing fodder. The half lethal dose ($LD_{50}$) of Se-enriched DB-yeast (9260.0 mg/kg body weight (BW), 18.97 mg/kg of Se content, non-toxic level) was considerably higher than that of $Na_2SeO_3$ (20.0 mg/kg BW, 5.08 mg/kg of Se content, highly toxic level) against mouse. Therefore, Se-enriched yeast prepared by re-using discarded *S. cerevisiae* from beer industry fermentation accumulation has the potential to be a safe and effective Se-enriched fodder additive.

**Keywords:** discarded *Saccharomyces cerevisiae*; se-enriched yeast; Se-supplemented fodder application; fermentation condition optimization

---

## 1. Introduction

Selenium (Se) is recognized as a nutritional trace element that is essential for the proper functioning of humans and animals with antioxidant functions. The recommended dietary dose of Se for humans is 55 μg per day [1,2]. This trace element is a component of a number of important selenoproteins, as well as enzymes including glutathione peroxidases (GSH-Px), iodothyronine deiodinases, selenophosphate synthetases, thioredoxin reductases, etc., which play critical roles in reproduction, thyroid hormone

metabolism, DNA synthesis, and protection from oxidative damage and infection [3–6]. There are approximately 0.5 to 1.0 billion people in 40 countries around the world facing potential adverse health impacts because of Se deficiency [7]. Se deficiency is associated with the loss of hair pigment and macrocytosis in intravenously fed children, and is the cause of other human diseases, such as Keshan disease (an endemic cardiomyopathy) and Kashin–Beck disease (a type of osteoarthritis) in China [8,9]. There is increasing evidence that Se-enriched food can prevent these Se-deficiency diseases and also provide protection against various forms of cancers [1,5,10,11]. Although Se normally enters the food chain from plants as they absorb inorganic Se from the soil and convert it into organic Se, the amount of naturally obtained Se is often insufficient for the optimal growth and development of animals in Se-deficient regions [12]. Therefore, it is essential to supplement human and animal diets with Se-enriched foods [2,12].

Available Se supplements include inorganic forms (i.e., sodium selenite ($Na_2SeO_3$), sodium hydrogen selenite ($NaHSeO_3$), and sodium selenate ($Na_2SeO_4$)), which are soluble but poorly bioavailable and potentially toxic. However, through bioconversion, inorganic Se is transformed into organic forms (i.e., Se-enriched yeast (Se-yeast) and the seleno-amino acid, L-selenomethionine (Se-Met)), which have decreased toxicity and elevated absorption rates compared with inorganic Se [13–16]. Se supplementation produced by microorganisms, especially yeast (i.e., *Saccharomyces cerevisiae*), received much attention in the past decade. Under appropriate conditions, yeast can utilize soluble sugars and organic acids to produce high-protein biomass while accumulating and incorporating large amounts of Se into organic Se-containing compounds, the main form of which is SeMet, which is the best Se source for organisms [13,17,18]. Therefore, since the Food and Drug Administration (FDA) approved the use of Se-yeast, an organic source of Se, in poultry diets in 2000, it became one of the most popular sources of Se supplementation in the agriculture and human nutritional supplement industries [15,19].

Grains, breads, meat, poultry, eggs, cereals, and fish are considered to be the major sources of Se in the diet [5]. In China, genetically improved farmed tilapia (*Oreochromis niloticus*) (GIFT) is a commercially important farmed freshwater fish because of its omnivorous feeding habit, rapid growth rate, and high fillet yield [20]. It can also be considered as an Se carrier in the human diet in some areas of Se deficiency since it is delicious and the total production of tilapia is 1.6 million tons in China, while the total global production is 5.0 million tons [21].

The abundant discarded beer yeast (*S. cerevisiae*) (DB-yeast) originating from the beer industry is a serious source of pollution; however, if DB-yeast is recycled and reused appropriately, it can be converted to a valuable social resource. Generally, DB-yeast from the beer industry is used as a high-protein fodder additive after simple drying. In this study, to improve its utilization value, inorganic Se ($Na_2SeO_3$) was bio-transformed to organic Se (Se-enriched DB-yeast) by re-using DB-yeast. The effects of the fermentation culture conditions on the bioaccumulation of Se in DB-yeast were investigated using a single-factor test and an $L_9(3^4)$-orthogonal test in Se-enriched media. The as-prepared Se-enriched DB-yeast was further used as the Se-enriched fodder additive for GIFT. The aim of this study was to develop a safe and effective Se-enriched fodder additive for Se-enriched food production by re-using DB-yeast accumulated from fermentation.

## 2. Materials and Methods

### 2.1. Strain, Media, and Growth Conditions

DB-yeast was obtained from Jinwei Beer Group Co. LTD (Guangzhou, China) (approximately 80% moisture). Malt extract (9.0% Brix) was obtained from Budweiser Beer Co. LTD (Foshan, China). For optimization of laboratorial fermentation conditions, 50.0 mL of fermentation medium was placed into a 250-mL conical flask. The preliminary laboratorial fermentation conditions were 30 °C, pH 5.0, 9.0% Brix, 9.0% inoculation volume (4.5 mL of DB-yeast), 10 µg·mL$^{-1}$ Se (22.35 µg/mL $Na_2SeO_3$, purity 98.0%), and 9 h of Se adding time. The mixture was cultured at 160 rpm for 24 h.

## 2.2. Fermentation Condition Optimization

Based on the primary fermentation primary conditions, a single-factor test and an $L_9(3^4)$-orthogonal test were conducted using biomass and total Se content as the indicators. For the single-factor test, temperature, pH, culture time, inoculation volume, Brix, Se concentration, and Se adding time were selected as single factors. Optimization of Se concentration (10.0, 20.0, and 30.0 µg/mL) (corresponding to 22.35, 44.70, and 67.05 µg/mL $Na_2SeO_3$, purity 98.0%), pH (3.0, 4.0, and 5.0), inoculation volume (9.0%, 12.0%, and 15.0%), and Se adding time (6, 9, and 12 h) were used for the $L_9(3^4)$-orthogonal test (Tables S1–S3, Supplementary Materials).

## 2.3. Determination of Total Biomass and Se Content of Se-Enriched DB-Yeast

A total of 50.0 mL of the culture medium was centrifuged at 4000 rpm for 10 min and washed thrice with deionized water. The obtained precipitate was dried at 105 °C using a vacuum drying oven for 4 h to obtain dry cells for biomass testing [22].

$$\text{Biomass (g/L)} = \frac{\text{Dried yeast (g)}}{\text{Volume of fermentation liquor (mL)}} \times 1000. \tag{1}$$

The total Se content of the dried sample was digested and then evaluated using a double-channel atomic fluorescence spectrophotometer AFS-2000 (Beijing Ke Chuang Hai Guang Instrument Co. LTD, Beijing, China) [22,23].

$$\text{Total Se content (µg/L)} = \frac{\text{Measured Se content (ng/mL)} \times \text{Dilution times}}{\text{Dried yeast (g)}} \times 1000. \tag{2}$$

## 2.4. Determination of Amino-Acid Composition

The amino-acid composition of the samples was determined at the Guangdong Detection Center of Microbiology (Guangdong, China) according to GBT18246-2000 and GB/T 6432-1994/7.2. Briefly, the dried sample was digested in a hydrolyzing tube containing 6 mol/L HCl by using a constant-temperature vacuum drying oven DHG-9076A (Wuxi Hualibang Machinery Technology Co. LTD, China) at 110 °C. The digested liquid was evaluated using an ammonia automatic analyzer L-8900 (Hitachi, Japan) after filtration. The admissible error for detection was as follows: the relative deviation of the measured values of parallel samples had to be ≤5% when the detected amino-acid content was ≤0.5%; the relative deviation of the measured values of parallel samples had to be ≤4% when the detected amino-acid content was >0.5%.

## 2.5. Determination of the Se-Met Contents of DB-Yeast and Se-Enriched DB-Yeast

Se-Met contents of DB-yeast and Se-enriched DB-yeast were determined by Qingdao Kechuang Quality Testing Co. LTD (Qingdao, China). Se-Met was purchased from Shanghai McLean Biochemical Technology Co., LTD (Shanghai, China). Briefly, 0.20 g of sample was mixed with 5 mL of 30 mmol/L Tris-HCl (pH 7.0) and 20 mg of pepsase XIV and incubated at 37 °C for 24 h. After centrifugation at 10,000 rpm for 10 min, the supernatant was passed through a 0.45-µm filter (Millipore, Shanghai, China) and then diluted 20 times. A total of 5 µL of the diluted solution was analyzed using LC–MS (Agilent1290-6460, USA) equipped with a Waters T3 chromatographic column (2.1 mm × 100 mm × 3 µm). Se-Met was detected at 290 nm and a column temperature of 25 °C using a mobile phase composed of a mixture of 0.1% methanoic acid aqueous solution and acetonitrile (95:5 *v/v*, 0–0.5 min; 50:50 *v/v*, 0.5–5 min; 0:100 *v/v*, 5–8 min) that was pumped through the column at a flow rate of 0.3 mL/min. The eluates were monitored by a mass spectrometer. The retention time and standard curve for Se-Met was 2.11 min (peak area = 545.05 × concentration − 101.03; $R^2$ = 1.000).

### 2.6. Characterization of Extracted Proteins from DB-Yeast and Se-Enriched DB-Yeast

#### 2.6.1. Protein Extraction

The dried Se-enriched yeast was stirred in 0.25 mol/L NaOH for 1 h at 65 °C. Then, the hydrolyzed liquid was centrifuged for 15 min at 4000 rpm to obtain the supernatant containing the extracted protein. The concentration of extracted protein was measured by Coomassie Brilliant Blue [24]. For the protein extraction, the extracted protein was diluted to 5 mg/mL and the pH was adjusted to 4.5. After stewing for 10 min and centrifugation at 4000 rpm for 15 min, the extracted protein was obtained as a precipitate.

#### 2.6.2. Effect of HCl on the pH of Extracted Proteins from DB-Yeast and Se-Enriched DB-Yeast

The pH of 20.0 mL of the protein solution (5 mg/mL and initial pH 10) was measured by a pH meter Phs-3G (Shanghai Thunder Magnetic Instrument, Shanghai, China) after adding 0, 1.0, 2.0, 3.0, 4.0, and 5.0 mL of 0.017 mol/L HCl.

#### 2.6.3. Determination of the Protein Reducing Power

The protein reducing power was determined according to the method reported by Shen et al. [25]. A total of 2.5 mL of protein sample (2.0, 4.0, 6.0, 8.0, and 10.0 mg/L) in 2.5 mL of 0.2 mol/L phosphate buffer (pH 6.6) was mixed with 2.5 mL of 1.0% potassium ferricyanide. The mixture was incubated at 50 °C for 20 min, and then 2.5 mL 10% trichloroacetic acid was added to terminate the reaction. After centrifugation at 4000 rpm for 10 min, 1.0 mL of supernatant was mixed with 0.2 mL of 0.1% ferric chloride for 10 min. The absorbance was measured at 700 nm, and an increased absorbance of the reaction mixture indicates an increased reducing power.

#### 2.6.4. Determination of Hydroxyl Radicals

The hydroxyl radicals of the proteins were measured using a hydroxyl radical kit (Nanjing Xinfan Biotechnology Co., LTD, Nanjing, China).

#### 2.6.5. Determination of the Total Amount of Sulfhydryl

The total amount of sulfhydryl of the proteins after $H_2O_2$ oxidation was measured using a BestBio total sulfhydryl detection kit (Shanghai BestBio Biotechnology Co., LTD, Shanghai, China).

#### 2.6.6. Determination of Protein Secondary Structures

Protein secondary structures were measured using a Fourier=transform infrared spectrometer Nexus 670 (Nicolet, WI, USA). The amide I region (1600–1700 $cm^{-1}$) of the FTIR spectrograms was analyzed by PeakFit 4.12 software using a Gaussian function second-derivative fitting. The relative percentages of protein secondary structures were calculated according to the corresponding region area [26,27].

### 2.7. The Half Lethal Dose (LD$_{50}$) Bioassay

The half lethal dose (LD$_{50}$) was measured at the GuangZhou Center for Disease Control and Prevention. According to the Horn method [28], $Na_2SeO_3$ fodder (4.46, 10.0, 21.5, and 46.4 mg/kg of Se content) and Se-enriched yeast fodder (2150, 4640, 10,000, and 21,500 mg/kg of Se content) were evaluated using SPF (specific pathogen-free) NIH (National Institutes of Health) female and male mice (Guangdong Medical Experimental Center, Guangdong, China). The mice were housed at a temperature of 25 ± 2 °C, 55.0 ± 10.0% humidity, with an alternating 12-h day and 12-h night cycle and free access to drinking water. They were fed 0.2 mL $Na_2SeO_3$ fodder or Se-enriched yeast fodder per 10.0 g body weight (BW) via gastric feeding after fasting for 6 h, with free access to drinking water.

The mice were observed after 24 h for signs of toxicity or lethality. Each experimental group contained three female and three male mice.

### 2.8. Fodder Bioassay for Mice

Kunming female mice KM (22 days, Guangdong Medical Experimental Center, Guangdong, China) were used for the Se-containing ($Na_2SeO_3$ or Se-enriched yeast) fodder bioassay. Mouse basic fodder (Guangdong Medical Experimental Center, Guangdong, China) was rod-like and designed based on the American Academy of Nutrition Standards AIN93. The nutrient content included 17.8% protein, 64.3% carbohydrate, 7.0% fat, 6.6% moisture, 4.17% ash, and 3766 kcal/kg. $Na_2SeO_3$ and Se-enriched yeast fodder both contained 0.2 mg/kg Se, and fodder without Se was used as the control. The feeding conditions were as follows: temperature $25 \pm 2$ °C, humidity $55.0 \pm 10.0$%, alternating 12-h day and 12-h night with free access to drinking water, weighing at 8:00 a.m., and 0.2 mL gastric feeding at 9:00 a.m. The experiment lasted 21 days, and feeding was stopped one day before the end of the feeding. Mouse eyeballs were extracted after diethyl ether anesthesia and immediately put into 5 mL of heparin sodium (100 U/mL). After freezing and centrifugation at 4 °C and 3000 rpm for 15 min, a faint yellow supernatant serum was obtained. Serum indexes were measured using a fully automatic biochemical analyzer (IDEXX, Maine, USA). After serum sampling, mice were executed via neck sudden death. The kidneys, liver, leg muscles, and heart were aseptically removed, oven-dried at 60 °C, and processed for the Se content assay. Each group contained three mice.

### 2.9. The Fodder Bioassay for Genetically Improved Farmed Tilapia (GIFT)

GIFT (Huadu Fish Breeding Base, China) were used for the Se-containing ($Na_2SeO_3$ or Se-enriched yeast) fodder bioassay. The GIFT basic fodder contained 5.0% fish meal, 30.0% bean pulp, 36.0% rapeseed meal, 24% flour, 2.0% soya-bean oil, 2.0% mixed fish premix, and 1.0% $CaH_2PO_4$. The amounts of $Na_2SeO_3$ and Se-enriched yeast added to the fodder were both 0.2 mg/kg, and fodder without Se was used as the control. After 60 days of feeding, the serum indexes and Se content of the gill, intestine, spleen, liver, and serum were analyzed. Serum indexes were measured using a fully automatic biochemical analyzer (IDEXX, Maine, USA). Each group contained 30 GIFT.

### 2.10. Scanning Electron Microscopy (SEM) and Energy-Dispersive Spectrometry (EDS)

Scanning electron microscopy (SEM) and energy-dispersive spectrometry (EDS) of brewing industrial DB-yeast and Se-enriched DB-yeast were carried out using a scanning electron microscope EVO 18 (Zeiss, Oberkochen, Germany).

### 2.11. Data Analysis

Data presented are the averages of at least three assays. Statistical analysis was carried out using SPSS 17.0 statistical software. The statistical significance was evaluated using the Student's *t*-test and a *p*-value <0.05 was considered statistically significant.

## 3. Results

### 3.1. The Optimization of the Fermentation Culture Conditions

Under the primary fermentation conditions, the Se content and biomass of the Se-enriched DB-yeast were 6.91 mg/L and 7.1 g/L, respectively. A single-factor experiment and $L_9(3^4)$-orthogonal test for optimization of fermentation conditions were used to improve the Se-enriched capacity of the DB-yeast.

Single-factor experiments for optimization of the fermentation conditions are shown in Figures 1 and 2. As shown in Figure 1A, both the Se content and biomass of the Se-enriched DB-yeast firstly increased and then decreased as the temperature increased from 24 to 32 °C. The optimal temperature was 28 °C. As shown in Figure 1B, as the initial pH increased (from 3.0 to 7.0), the biomass of the Se-enriched DB-yeast continuously increased; however, the Se content of the Se-enriched

DB-yeast firstly increased and then decreased as initial pH increased. These data indicate that a slightly acidic environment is good for industrial discarded *S. cerevisiae* growth and Se fermentation accumulation. The optimal initial pH was 4.0 under overall consideration. As shown in Figure 1C, the biomass of the Se-enriched DB-yeast slightly decreased but, overall, it was maintained at a high level over a culture time of 24 to 32 h. The Se content of the Se-enriched DB-yeast firstly increased and then decreased as the cultured time increased. The optimal cultured time was 30 h. As shown in Figure 1D, the biomass of the Se-enriched DB-yeast increased as the inoculation volume increased from 4.0% to 15.0%. The Se content of the Se-enriched DB-yeast firstly increased and then decreased, and the optimal inoculation volume was 12.0%. As shown in Figure 2A, the Se content of the Se-enriched DB-yeast obviously increased as the concentration of added Se increased from 10 to 50 µg/mL. However, the biomass of the Se-enriched DB-yeast showed the opposite trend, which was probably related to the cytotoxicity of inorganic Se as its concentration increased. The optimal Se adding concentration was 30 µg/mL under overall consideration. As shown in Figure 2B, the biomass of the Se-enriched DB-yeast increased as the time of Se addition increased. The Se content of the Se-enriched DB-yeast firstly increased and then decreased, and the optimal Se addition time was 9 h.

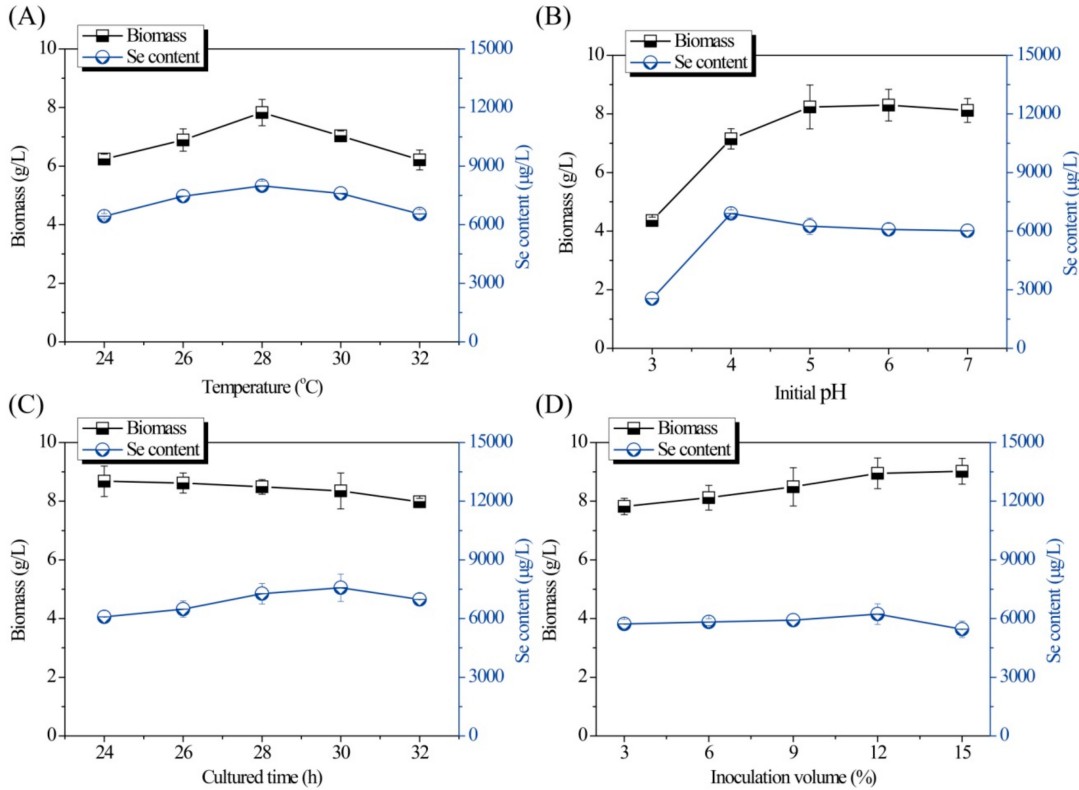

**Figure 1.** Effects of (**A**) temperature, (**B**) pH, (**C**) culture time, and (**D**) inoculation volume on Se content and biomass of Se-enriched discarded beer (DB)-yeast fermentation.

To further improve the fermentation conditions, the initial pH (3.0, 4.0, and 5.0), inoculation volume (9.0%, 12.0%, and 15.0%), added Se (10, 20, and 30 µg/mL), and Se addition time (6, 9, and 12 h) were used for the $L_9(3^4)$-orthogonal test. The parameter design, extreme difference analysis, and variance analysis results are shown in Tables S1–S3 (Supplementary Materials). The computed *F*-value of Se added (A) was much greater than that of the tabular *F0.01* value, suggesting that it has an extremely significant effect on the Se content of the Se-enriched DB-yeast. In addition, the computed *F*-value of the Se addition time (D) was between the tabular *F0.01* and *F0.05* value, suggesting that it has a significant effect on the Se content of the Se-enriched DB-yeast. By contrast, the effects of pH (B) and inoculation volume (C) on the Se content of the Se-enriched DB-yeast were not significant.

Hence, the effect sizes of the significant variables were as follows: added Se (A) > Se addition time (D) > inoculation volume (C) > pH (B). The combination of the optimization conditions was $A_3B_1C_3D_2$.

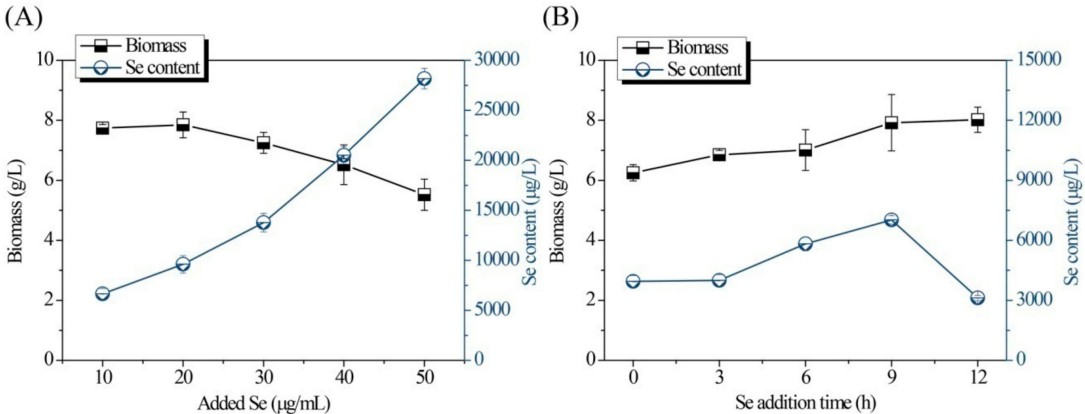

**Figure 2.** Effects of (**A**) Se concentration and (**B**) Se addition time on Se content and biomass of Se-enriched DB-yeast fermentation.

Since the initial pH of the medium was 5.0 and the effect of the pH was not significant, the components of the fermentation conditions were chosen as follows: 28 °C, pH 5.0, 9.0% Brix, 15.0% inoculation volume, 30 μg/mL added Se, and 9 h of Se addition time with culturing at 160 rpm for 30 h. To confirm the prediction, another experiment was carried out under the optimum fermentation conditions. The biomass of the optimized Se-enriched DB-yeast was 7.3 g/L, which was only slightly increased compared with that of the primary fermentation conditions. Remarkably, the Se content of the optimized Se-enriched DB-yeast reached 14.95 mg/L, which was a 2.16-fold increase compared with the original value of 6.91 mg/L.

*3.2. Characterization of the Se-Enriched DB-Yeast*

To better understand the differences of DB-yeast and Se-enriched DB-yeast, SEM and EDS were performed to investigate the surface morphology and Se element content (Figure 3). As shown in Figure 3A, DB-yeast was yellow (Figure 3A, inset) and spherical. As shown in Figure 3B, Se-enriched DB-yeast was sandy beige (Figure 3B, inset) and spherical but the surface was surrounded with some debris. As shown in Figure 3C, the Se element content of DB-yeast was only 0.02%. As shown in Figure 3D, the Se element content of Se-enriched DB-yeast was 0.54%, indicating that Se was successfully accumulated by DB-yeast fermentation. The Se-Met content of the DB-yeast and Se-enriched DB-yeast was further analyzed by LC–MS (Figure 4). The results showed that Se-Met was not detected in DB-yeast (<0.05 mg/kg) (Figure 4B), whereas it was found in Se-enriched DB-yeast (up to 1.23 ± 0.02 mg/kg) (Figure 4C), which proved that Se-enriched yeast was obtained after DB-yeast fermentation.

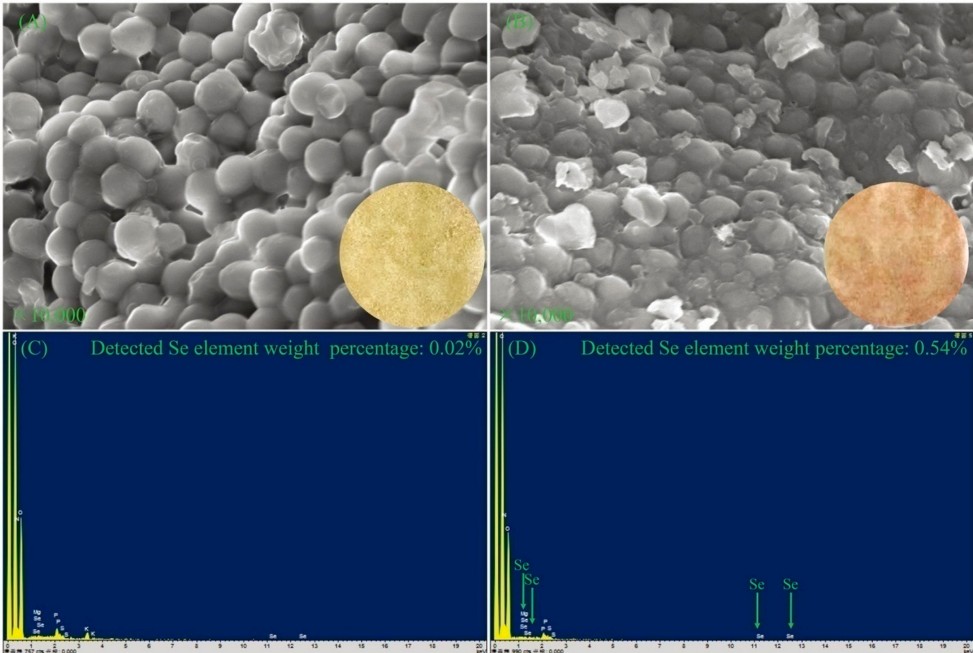

**Figure 3.** SEM and energy-dispersive spectrometry (EDS) for DB-yeast and Se-enriched DB-yeast. (**A**) SEM of DB-yeast. The inset image is powdery DB-yeast (yellow). (**B**) SEM of Se-enriched DB-yeast. The inset image is powdery Se-enriched DB-yeast (sandy beige). (**C**) EDS of DB-yeast. (**D**) EDS of Se-enriched DB-yeast.

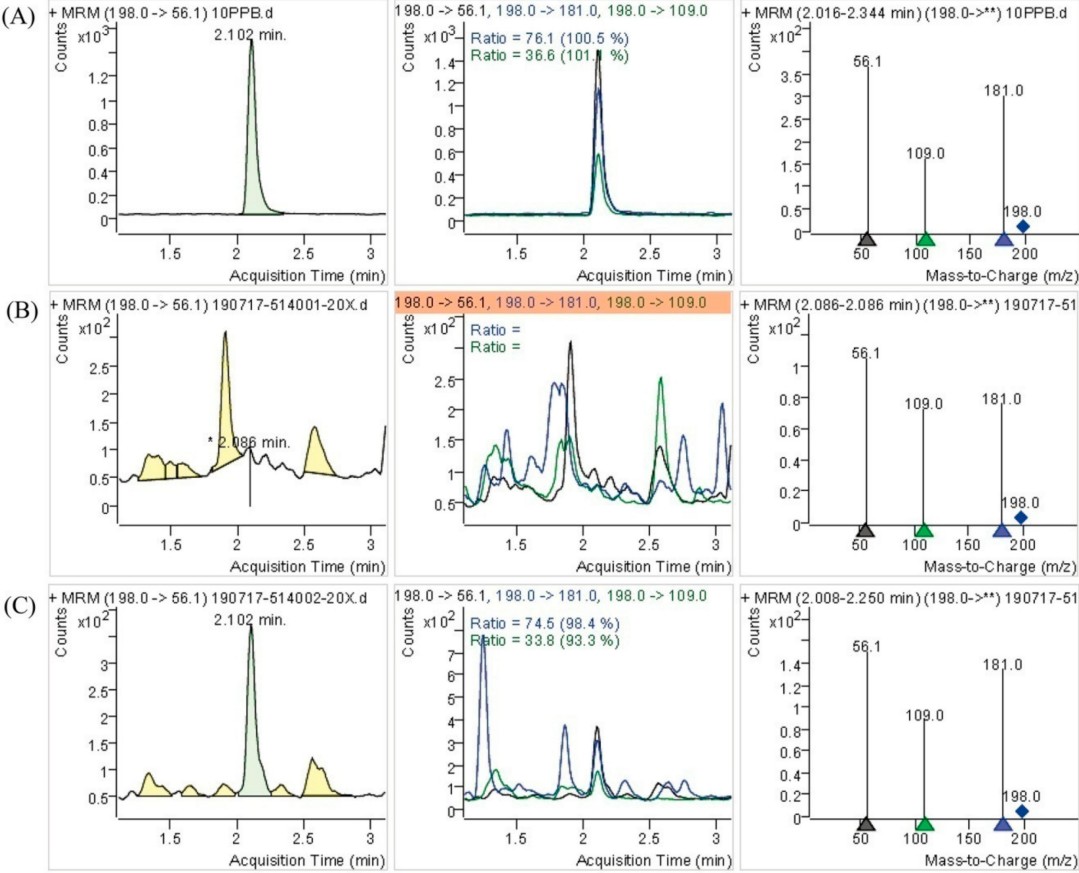

**Figure 4.** Determination of ʟ-selenomethionine (Se-Met) of DB-yeast and Se-enriched DB-yeast by LC–MS. (**A**) Se-Met standard substance (10 mg/L). (**B**) DB-yeast. (**C**) Se-enriched DB-yeast.

The effects of Se on the amino-acid composition of DB-yeast and Se-enriched DB-yeast were also analyzed (Table 1). The total amino-acid content of Se-enriched DB-yeast (43.3 g/100g) was 9.9% higher than DB-yeast (39.4 g/100 g). For the relative ratio of each amino acid, 11 kinds of amino acids, including the alkaline amino acids Arg, Lys, and His, were increased by 1.59%, 0.67%, and 0.11%, respectively, whereas, six kinds of the amino acids, including acidic amino acids Glu and Asp and the sulfur-containing amino acids Cys and Met, were decreased by 3.85%, 0.07%, 0.35%, and 0.07%, respectively, from Se-enriched DB-yeast compared with DB-yeast.

**Table 1.** Amino-acid composition of discarded beer (DB)-yeast and Se-enriched DB-yeast.

| Amino Acids | DB-Yeast | | Se-Enriched DB-Yeast Yeast | | Relative Ratio Difference |
| --- | --- | --- | --- | --- | --- |
| | Content [a] | Ratio [b] | Content [a] | Ratio [b] | |
| Total | 39.4 | - | 43.3 | - | - |
| Arg | 2.23 | 5.7% | 3.14 | 7.3% | 1.59% |
| Pro | 1.36 | 3.5% | 1.8 | 4.2% | 0.71% |
| Lys | 3.44 | 8.7% | 4.07 | 9.4% | 0.67% |
| Gly | 2.25 | 5.7% | 2.76 | 6.4% | 0.66% |
| Ser | 2.14 | 5.4% | 2.48 | 5.7% | 0.30% |
| Val | 1.48 | 3.8% | 1.69 | 3.9% | 0.15% |
| Tyr | 1.4 | 3.6% | 1.59 | 3.7% | 0.12% |
| His | 0.95 | 2.4% | 1.09 | 2.5% | 0.11% |
| Thr | 2.28 | 5.8% | 2.54 | 5.9% | 0.08% |
| Leu | 3.14 | 8.0% | 3.48 | 8.0% | 0.07% |
| Ala | 2.65 | 6.7% | 2.93 | 6.8% | 0.04% |
| Phe | 1.88 | 4.8% | 2.04 | 4.7% | -0.06% |
| Asp | 4.45 | 11.3% | 4.86 | 11.2% | -0.07% |
| Met | 0.53 | 1.3% | 0.55 | 1.3% | -0.07% |
| Ile | 2.14 | 5.4% | 2.24 | 5.2% | -0.26% |
| Cys | 0.4 | 1.0% | 0.29 | 0.7% | -0.35% |
| Glu | 6.73 | 17.1% | 5.73 | 13.2% | -3.85% |

[a] g/100g; [b] ratio compared with total amino-acid content.

### 3.3. Characterization of Extracted Proteins from DB-Yeast and Se-Enriched DB-Yeast

The pH of the solution treated with HCl, and the reducing power, hydroxyl radical removal capacity, and total sulfhydryl and secondary structure composition treated with different volumes of $H_2O_2$ of extracted proteins from DB-yeast and Se-enriched DB-yeast were evaluated to better understand the excellent characteristics of Se-enriched DB-yeast. The pH of the Se-enriched DB-yeast extracted protein solution (from $10.00 \pm 0.06$ to $5.25 \pm 0.07$) decreased more slowly than that of the DB-yeast extracted protein solution (from $10.02 \pm 0.04$ to $2.23 \pm 0.07$) following the addition of HCl (Figure 5A). The reducing power of the Se-enriched DB-yeast extracted protein ($1.65 \pm 0.08$) was significantly higher than that of the DB-yeast extracted protein ($1.20 \pm 0.07$) (*p*-value = 0.002) (Figure 5B). The hydroxyl radical removal capacity of the Se-enriched DB-yeast extracted protein ($53.34 \pm 4.56\%$) was significantly higher than that of the DB-yeast extracted protein ($81.61 \pm 5.62\%$) (*p*-value = 0.002) (Figure 5C). The sulfhydryl content of the Se-enriched DB-yeast extracted protein (from $37.5 \pm 0.7$ μmol/L to $28.7 \pm 0.9$ μmol/L) decreased more slowly than that of the DB-yeast extracted protein (from $38.2 \pm 1.1$ μmol/L to $25.9 \pm 0.8$ μmol/L) as the $H_2O_2$ concentration increased (from 0 to 16 mmol/L) (Figure 5D). As shown in Table 2, after treatment with $H_2O_2$, the β-sheet content decreased, whereas the β-turn, α-helix, and random coil content increased. The β-sheet content of the Se-enriched DB-yeast extracted protein (from 47.55% to 43.08%) decreased more slowly than that of the DB-yeast extracted protein (from 47.12% to 40.19%) as the $H_2O_2$ concentration increased (from 0 to 16 mmol/L). In addition, as the most abundant structure, β-sheet was more susceptible to oxidation

and denaturation than the other structures. Therefore, Se can protect Se-enriched DB-yeast extracted protein from oxidation and denaturation.

**Table 2.** The secondary structure composition of extracted proteins from DB-yeast and Se-enriched DB-yeast after treatment with $H_2O_2$.

| $H_2O_2$ [a] | DB-Yeast Extracted Protein [b] | | | | Se-enriched DB-Yeast Extracted Protein [b] | | | |
|---|---|---|---|---|---|---|---|---|
| | β-Sheet | β-Turn | α-Helix | Random Coil | β-Sheet | β-Turn | α-Helix | Random Coil |
| 0 | 47.12 | 17.67 | 11.54 | 23.67 | 47.55 | 17.22 | 11.51 | 23.72 |
| 0.05 | 46.46 | 17.81 | 11.75 | 23.98 | 46.89 | 17.54 | 11.67 | 23.90 |
| 0.25 | 45.97 | 17.89 | 11.92 | 24.22 | 46.15 | 17.75 | 11.89 | 24.21 |
| 1 | 44.71 | 18.13 | 12.17 | 24.99 | 45.23 | 18.05 | 12.13 | 24.59 |
| 4 | 42.88 | 18.54 | 12.79 | 25.79 | 44.37 | 18.24 | 12.45 | 24.94 |
| 16 | 40.19 | 19.06 | 13.23 | 27.52 | 43.08 | 18.34 | 12.86 | 25.72 |

[a] mM; [b] secondary structure composition (%).

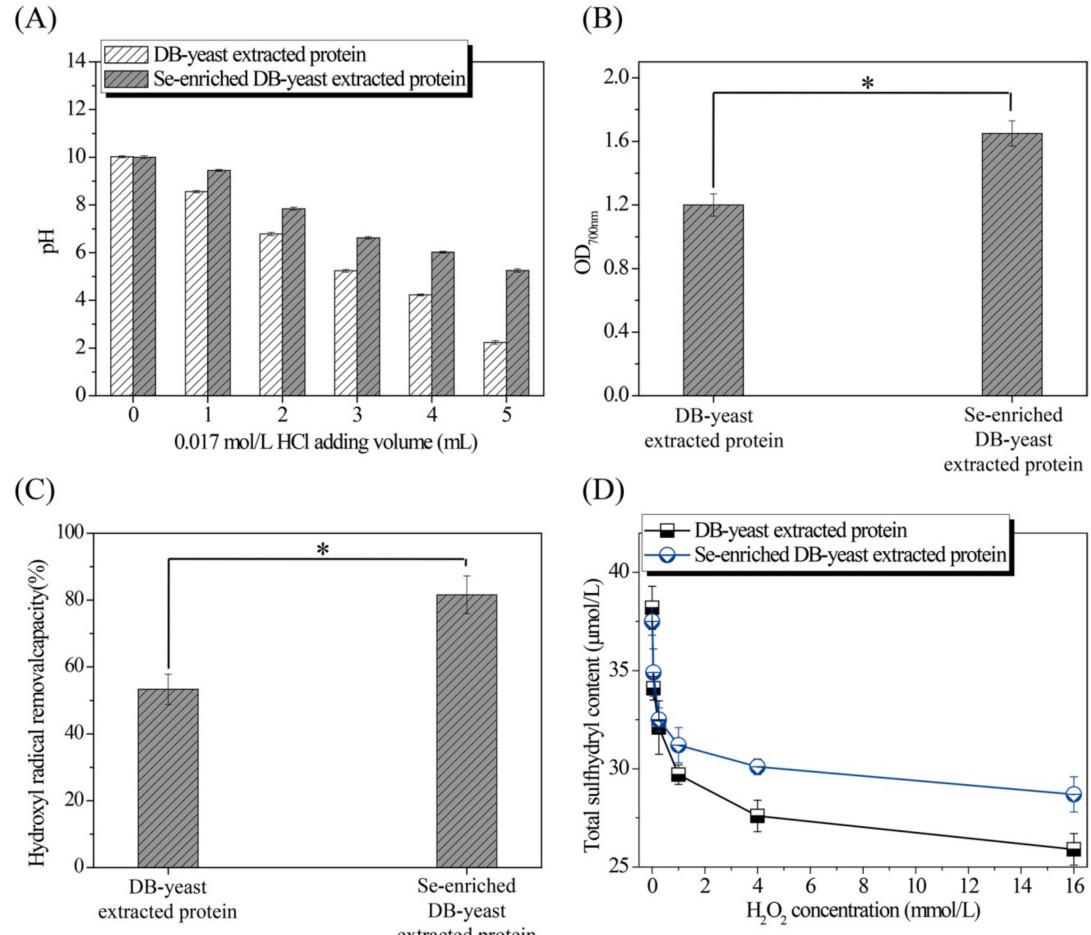

**Figure 5.** Characterization of DB-yeast and Se-enriched DB-yeast extracted proteins. (**A**) pH of extracted proteins (20 mL) after mixing with 0.017 mol/L HCl (0 to 5 mL). (**B**) Reducing power of extracted proteins (10.0 mg/mL of extracted proteins). (**C**) Hydroxyl radical removal capacity of extracted proteins (0.4 mg/mL of extracted proteins). (**D**) Effect of different $H_2O_2$ concentrations on total sulfhydryl content (3.6 mg/L of extracted proteins). * Statistically significant (*p*-value <0.05).

### 3.4. $LD_{50}$ Bioassay of Se-Enriched DB-Yeast

To understand the toxicity of Se-enriched DB-yeast as a fodder additive, $Na_2SeO_3$ and Se-enriched DB-yeast were evaluated by the Horn method using mouse as the biological indicator. As shown in Table 3, the $LD_{50}$ of $Na_2SeO_3$ was 20.0 mg/kg BW (5.08 mg/kg of Se content, 13.7–21.9 mg/kg BW at 95% confidence bounds), which was identified as a highly toxic level. Surprisingly, after fermentation accumulation by DB-yeast, the $LD_{50}$ of Se-enriched DB-yeast was 9260.0 mg/kg BW (18.97 mg/kg of Se content, 6360.0–13,500.0 mg/kg BW at 95% confidence bounds), which was identified as a non-toxic level.

**Table 3.** Evaluation of $Na_2SeO_3$ and Se-enriched DB-yeast toxicity against mouse by using Horn method. $LD_{50}$—half lethal dose.

| Se Resource | $LD_{50}$ [a] | 95% Confidence Bounds [a] |
|---|---|---|
| $Na_2SeO_3$ | 20.0 | 13.7–21.9 |
| Se-enriched DB-yeast | 9260.0 | 6360.0–13,500.0 |

[a] mg/kg body weight (BW).

### 3.5. Fodder Bioassays of $Na_2SeO_3$ and Se-Enriched DB-Yeast as the Fodder Additive for Mouse and GIFT

For mouse fodder bioassays, the effects of different Se sources, including $Na_2SeO_3$ (0.2 mg/kg Se) and Se-enriched DB-yeast (0.2 mg/kg Se), on mouse growth after 21 days of feeding are shown in Table 4. After 21 days of feeding, the weights of all treated mice increased (the weight gain rates of feeding fodders with no Se additive, $Na_2SeO_3$ additive, and Se-enriched DB-yeast additive were 35.42 ± 1.81%, 40.12 ± 2.07%, and 40.62 ± 1.78%, respectively). The weight gain rates of the $Na_2SeO_3$ additive group and Se-enriched DB-yeast additive group were obviously (approximately 5.0%) higher than that of the no Se-additive group; however, there was no difference between them. Interestingly, the serum GSH-Px of both the $Na_2SeO_3$ additive group (280.72 ± 8.82 U, increased approximately 23.1%) and the Se-enriched DB-yeast additive group (293.05 ± 9.63 U, increased approximately 28.5%) was significantly higher than that of the no Se-additive group (228.11 ± 9.27 U). Surprisingly, compared with the $Na_2SeO_3$ additive group, the serum GSH-Px of the Se-enriched DB-yeast additive group was only slightly increased (approximately 4.4%). Not surprisingly, compared with the no Se-additive control, both the $Na_2SeO_3$ and Se-enriched DB-yeast could significantly promote Se sedimentation for all mouse organs including liver, serum, intestine, spleen, and muscle. Remarkably, the Se sedimentation contents of all mouse organs from the Se-enriched DB-yeast additive group were significantly higher than those of the $Na_2SeO_3$ additive group. The Se sedimentation content of the $Na_2SeO_3$ additive group was approximately 2.9 times (1.46 ± 0.21 mg/kg), 2.6 times (2.53 ± 0.18 mg/kg), 2.0 times (0.79 ± 0.06 mg/kg), 2.5 times (5.16 ± 0.52 mg/kg), and 3.1 times (0.34 ± 0.02 mg/kg) higher than the no Se-additive group for liver, serum, intestine, spleen, and muscle, respectively. In addition, the Se sedimentation content of the Se-enriched DB-yeast additive group was approximately 3.8 times (5.55 ± 0.56 mg/kg), 1.4 times (3.47 ± 0.26 mg/kg), 4.7 times (3.74 ± 0.34 mg/kg), 2.0 times (10.47 ± 0.79 mg/kg), and 1.4 times (0.46 ± 0.05 mg/kg) higher than the $Na_2SeO_3$ additive group for liver, serum, intestine, spleen, and muscle, respectively.

For GIFT fodder bioassays, the effects of different Se sources, including $Na_2SeO_3$ (0.6 mg/kg Se) and Se-enriched DB-yeast (0.6 mg/kg Se), on GIFT growth after 60 days feeding are shown in Table 5. After 60 days of feeding, the average body weight in each experimental group increased and no deaths occurred. The weights of the feeding fodders with no Se additive, $Na_2SeO_3$ additive, and Se-enriched DB-yeast additive increased by 202.46 ± 6.02%, 217.37 ± 2.23%, and 219.40 ± 2.84%, respectively. The weight gain rates of both Se-additive-containing groups, including the $Na_2SeO_3$ additive group and Se-enriched DB-yeast additive group, were obviously (approximately 15.9%) higher than that of the no Se-additive group; however, there was no significant difference between them. Interestingly, the serum GSH-Px of both the $Na_2SeO_3$ additive group (45.80 ± 2.0 U) and the

Se-enriched DB-yeast additive group (98.23 ± 3.4 U) was significantly higher than that of the no Se-additive group (39.50 ± 2.1 U). Remarkably, the serum GSH-Px of the Se-enriched DB-yeast additive group was significantly higher than that of the no Se-additive group (approximately 2.5 times) and the $Na_2SeO_3$ additive group (approximately 2.1 times).

As predicted, the Se sedimentation contents of all organs including liver, serum, intestine, spleen, and muscle from the Se-enriched DB-yeast additive group and the $Na_2SeO_3$ additive group were significantly higher than those of the no Se-additive group. In addition, the Se sedimentation contents of all mouse organs from the Se-enriched DB-yeast additive group were significantly higher than those of the $Na_2SeO_3$ additive group. The Se sedimentation contents of the $Na_2SeO_3$ additive group were approximately 1.6 times (10.26 ± 0.71 mg/kg), 1.2 times (3.41 ± 0.20 mg/kg), 1.7 times (5.42 ± 0.48 mg/kg), 1.4 times (7.65 ± 0.43 mg/kg), and 2.4 times (0.74 ± 0.05 mg/kg) higher than the no Se-additive group for liver, serum, intestine, spleen, and muscle, respectively. Additionally, the Se sedimentation content of the Se-enriched DB-yeast additive group was approximately 1.2 times (11.87 ± 0.54 mg/kg), 1.3 times (4.37 ± 0.23 mg/kg), 1.4 times (7.66 ± 0.36 mg/kg), 1.4 times (10.88 ± 0.69 mg/kg), and 1.8 times (1.34 ± 0.04 mg/kg) higher than the $Na_2SeO_3$ additive group for liver, serum, intestine, spleen, and muscle, respectively.

**Table 4.** Effects of the Se source on mouse growth after 21 days of feeding.

| Test Items | | No Se-Additive Control | $Na_2SeO_3$ Additive 0.2 mg/kg Se | Se-Enriched DB-Yeast Additive 0.2 mg/kg Se |
|---|---|---|---|---|
| Weight gain rate [d] | | 35.42 ± 1.81 | 40.12 ± 2.07 [a] | 40.62 ± 1.78 [b] |
| GSH-Px | | 228.11 ± 9.27 | 280.72 ± 8.82 [a] | 293.05 ± 9.63 [b] |
| Se sedimentation content [e] | Liver | 0.50 ± 0.07 | 1.46 ± 0.21 [a,c] | 5.55 ± 0.56 [b,c] |
| | Serum | 0.98 ± 0.07 | 2.53 ± 0.18 [a,c] | 3.47 ± 0.26 [b,c] |
| | Intestine | 0.39 ± 0.05 | 0.79 ± 0.06 [a,c] | 3.74 ± 0.34 [b,c] |
| | Spleen | 2.04 ± 0.16 | 5.16 ± 0.52 [a,c] | 10.47 ± 0.79 [b,c] |
| | Muscle | 0.11 ± 0.03 | 0.34 ± 0.02 [a,c] | 0.46 ± 0.05 [b,c] |

[a] Statistically significant for $Na_2SeO_3$-containing fodder compared with control ($p$-value < 0.05); [b] statistically significant for Se-enriched DB-yeast-containing fodder compared with control ($p$-value < 0.05); [c] statistically significant for Se-enriched DB-yeast-containing fodder compared with $Na_2SeO_3$-containing fodder ($p$-value < 0.05); [d] wt.%; [e] mg/kg; GSH-Px, glutathione peroxidase (U).

**Table 5.** Effects of the Se source on genetically improved farmed tilapia (*Oreochromis niloticus*) (GIFT) growth after 60 days of feeding.

| Test Items | | No Se-Additive Control | $Na_2SeO_3$ Additive 0.6 mg/kg Se | Se-Enriched DB-Yeast Additive 0.6 mg/kg Se |
|---|---|---|---|---|
| Weight gain rate [d] | | 202.46 ± 6.02 | 217.37 ± 2.23 [a] | 219.40 ± 2.84 [b] |
| Serum GSH-Px | | 39.50 ± 2.1 | 45.80 ± 2.0 [a,c] | 98.23 ± 3.4 [b,c] |
| Se sedimentation content [e] | Liver | 6.38 ± 0.26 | 10.26 ± 0.71 [a,c] | 11.87 ± 0.54 [b,c] |
| | Serum | 2.93 ± 0.17 | 3.41 ± 0.20 [a,c] | 4.37 ± 0.23 [b,c] |
| | Intestine | 3.22 ± 0.24 | 5.42 ± 0.48 [a,c] | 7.66 ± 0.36 [b,c] |
| | Spleen | 5.47 ± 0.55 | 7.65 ± 0.43 [a,c] | 10.88 ± 0.69 [b,c] |
| | Muscle | 0.31 ± 0.06 | 0.74 ± 0.05 [a,c] | 1.34 ± 0.04 [b,c] |

[a] Statistically significant for $Na_2SeO_3$-containing fodder compared with control ($p$-value < 0.05); [b] statistically significant for Se-enriched DB-yeast-containing fodder compared with control ($p$-value < 0.05); [c] statistically significant for Se-enriched DB-yeast-containing fodder compared with $Na_2SeO_3$-containing fodder ($p$-value < 0.05); [d] wt.%; [e] mg/kg; GSH-Px, glutathione peroxidase (U).

## 4. Discussion

At a high concentration, Se is toxic and affects the central nervous system; however, at a low concentration, Se is an essential microelement for humans since its prominent effects in disease and cancer prevention are well documented [2,5,8]. Se exists in nature in inorganic and organic forms. Both types of Se can prevent and treat various diseases caused by Se deficiency. However, organic

Se is less toxic and has a higher absorption rate than inorganic Se [14,15]. In this study, inorganic Se ($Na_2SeO_3$) was bio-transformed into organic Se (Se-enriched DB-yeast) through fermentation accumulation by re-using DB-yeast obtained from the beer industry for Se-enriched fodder application. The bioavailability and non-toxic characteristics of Se-enriched DB-yeast indicated that it has the potential to be used as a safe and effective Se-enriched fodder additive.

DB-yeast was chosen as the Se accumulation carrier to avoid the potential risk of pollution and to limit wasting of resources. However, DB-yeast is not a professional Se accumulation carrier; therefore, a single-factor experiment and an $L_9(3^4)$-orthogonal test for optimization of fermentation conditions were employed to improve its Se accumulation capacity before incorporating it as a fodder additive. To easily obtain the fermentation medium, beer industry fermentation medium was directly used for the initial fermentation medium for the DB-yeast. Interestingly, temperature, pH, and Brix of the initial fermentation condition did not significantly affect the Se accumulation capacity of the DB-yeast, indicating that it is easy to control the fermentation conditions by controlling the concentration of added Se, the time of Se addition, and inoculation volume. Furthermore, the concentration and timing of the added Se can significantly affect the Se accumulation capacity of *S. cerevisiae*. Yin et al. [13] reported that, after fermentation condition optimization, using a culture medium supplemented with 15.0 μg/mL $Na_2SeO_3$ added at 9 h after inoculation, which is during the logarithmic growth phase, the total Se content and the maximum biomass in a *S. cerevisiae* strain could reach 5.9 mg/L and 9.23 g/L, respectively. Yoshinaga et al. [29] reported one *S. cerevisiae* isolate with high levels of Se accumulation capacity (reaching 1.4 mg/g) incubated in an optimized Se accumulation medium (containing 25.2 μg/mL $Na_2SeO_3$) after directed evolution by glycerol- and selenium-containing medium selection. Therefore, after fermentation condition optimization (containing 30.0 μg/mL $Na_2SeO_3$ and an adding time of 9 h after inoculation), the Se content of the Se-enriched DB-yeast reached 14.95 mg/L (approximately 2.0 mg/g), which indicated a considerably high Se accumulation capacity. The maximum biomass of the Se-enriched DB-yeast was 7.3 g/L, which was relatively low. This may be attributed to variations in the degree of growth performance of the DB-yeast in the beer industry fermentation medium.

Through bioconversion, inorganic Se is transformed into organic Se such as selenomethionine or selenocysteine, possibly via replacement of the sulfur in methionine and cysteine, finally forming newly synthesized protein [5,30]. Therefore, compared with DB-yeast, the sulfur-containing amino-acid content of Se-enriched DB-yeast decreased. Interestingly, on the one hand, the total amino-acid content of the Se-enriched DB-yeast increased, which may be attributed to Se reducing the protein degradation resulting from oxidative damage; on the other hand, alkaline amino-acid contents increased, whereas acidic amino-acid content decreased, revealing the stress response of DB-yeast for $Se_2O_3^{2-}$.

Se is also a key component of numerous functional selenoproteins including five glutathione peroxidases (GSH-Px), two deiodinases, thioredoxin reductases, and selenophosphate synthetases [5]. These five GSH-Px can detoxify $H_2O_2$ and fatty-acid-derived hydroperoxides, thus contributing to the antioxidant defense against reactive molecules and free radicals, complementing the effects of vitamin E [10]. Therefore, compared with the DB-yeast extracted protein, the reducing capacity, hydroxyl radical removal capacity, and sulfhydryl content after treatment with $H_2O_2$ of the Se-enriched DB-yeast extracted protein were obviously increased. These increased antioxidant defense abilities may due to functional selenoprotein synthesis through inorganic Se bioconversion, indicating that Se-enriched DB-yeast can be used as the bioavailable organic Se resource for fodder Se-supplemented additive.

For the GIFT bioassay, the Se content of the fodder additives was set as 0.6 mg/kg Se, which was three times more than the mouse bioassay (0.2 mg/kg Se). The Se content increase was because the fish fodder was diluted by water in the GIFT bioassay. Ibrahim et al. [31] demonstrated that Se supplementation in broiler diets significantly improved weight gain, final body weight, and meat quality without increasing the feeding cost. Similarly, both Se sources, $Na_2SeO_3$ and Se-enriched DB-yeast, could improve mouse and GIFT growth performance. However, the growth performance of mice and GIFT treated with Se-enriched DB-yeast was not significantly increased compared with those treated with $Na_2SeO_3$, which was slightly different from the results reported by Yoon et al. [32],

where organic Se supplementation improved growth compared with inorganic Se supplementation in broiler chickens; however, this may because of variations in the growth performance of different species in the presence of Se. The GSH-Px activity level in the liver and serum is indicative of Se the status and antioxidant level of the organism [11,33]. GIFT treated with Se-enriched DB-yeast had significantly higher serum GSH-Px activities than those treated with $Na_2SeO_3$ or no Se-additive, which was attributed to the Se sedimentation content in the serum, indicating that Se-enriched DB-yeast can be used as an effective Se-enriched fodder additive for GIFT. Both mouse and GIFT had a higher Se sedimentation content of the liver, intestine, and spleen than that of muscle, indicating that the viscera, particularly the liver and spleen, were the major locations of Se sedimentation, similar to the results from Misra et al. [34] for *Oncorhynchus mykiss* treated with selenomethionine. After treatment with Se-enriched DB-yeast, the Se sedimentation content of the edible part of GIFT muscle reached $1.34 \pm 0.04$ mg/kg, which was 4.3 times and 1.8 times higher than the no Se-additive and $Na_2SeO_3$ groups, respectively, and also higher than most recommended food sources of Se such as tuna (approximately 1.08 mg/kg), yellowfin (approximately 1.08 mg/kg), halibut (approximately 0.55 mg/kg), and sardines (approximately 0.53 mg/kg) [5]. Therefore, GIFT treated with Se-enriched DB-yeast can be further used as an effective food source of Se in the human diet.

A previous study revealed that the minimum lethal dose of Se in the form of $Na_2SeO_3$ varies from 1.5 to 8.0 mg/kg body weight in farm animals [12]. Here, the $LD_{50}$ of Se-enriched DB-yeast (18.97 mg/kg of Se content) was considerably higher than that of $Na_2SeO_3$ (5.08 mg/kg of Se content) against mouse, indicating that organic Se (Se-enriched DB-yeast) is less toxic (non-toxic) than inorganic Se ($Na_2SeO_3$), similar to the results reported by Adadi et al. [11]. Therefore, Se-enriched DB-yeast has the potential to be a safe and effective Se-enriched fodder additive.

In the future, Se-enriched DB-yeast can also be applied as a fodder additive for pig, chicken, cattle, etc. in an effort to produce more types of Se-enriched foods as a way to combat Se deficiency. Moreover, Se-enriched beer prepared by adding organic Se during the beer fermentation process may also be a good source of Se-enriched food and warrants further study.

## 5. Conclusions

To the best of our knowledge, the present study demonstrated, for the first time, that inorganic Se ($Na_2SeO_3$) was bio-transformed to organic Se (Se-enriched DB-yeast) through fermentation accumulation by re-using brewing industrial discarded *S. cerevisiae* for Se-enriched fodder application. The Se content and biomass of Se-enriched DB-yeast reached 14.95 mg/L and 7.3 g/L, respectively, under the optimized conditions. The bioavailability and non-toxic characteristics of Se-enriched yeast prepared by re-using brewing industrial discarded *S. cerevisiae* fermentation accumulation suggest that it has the potential to be a safe and effective Se-enriched fodder additive.

**Supplementary Materials:** The following are available online at http://www.mdpi.com/2076-3417/9/18/3777/s1: Table S1: $L_9(3^4)$-orthogonal test for Se content of Se-enriched DB-yeast fermentation liquid; Table S2 Significance analysis of the factors in the $L_9(3^4)$-orthogonal test; Table S3 Analysis of variance for the selected factorial model.

**Author Contributions:** F.L., X.L., F.Z. and Z.F. performed the experiments. G.W. and X.S. conceptualized and directed the study. F.L. and G.W. drafted the manuscript. G.W. revised the manuscript.

**Funding:** This work was supported by the Guangdong Science and Technology Planning Project (No. 2016A010105023) and the Guangzhou Science and Technology Planning Project (No. 201704020055).

**Conflicts of Interest:** The authors declare no conflicts of interest.

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
