# Peer review of "Preparation of Selenium-Enriched Yeast by Re-Using Discarded Saccharomyces cerevisiae from the Beer Industry for Se-Supplemented Fodder Applications"

_applsci, doi:10.3390/app9183777_

Round 1

Reviewer 1 Report

The manuscript " Preparation of Selenium-enriched Yeast by re-using Discarded Saccharomyces 1 cerevisiae in Beer Industry for Se-Supplemented Fodder Application " investigates on the selenium-enriched yeast coming from beer industry.  The manuscript is well written and easy to follow. The results on the optimization of  the biotechnological process are of interest in view of a possible use in fodder application.  

However there are some concerns that should be fixed:

Strain, Media and Growth Conditions section should be more detailed.  What were the modalities of fermentation trials?? Flasks , Bioreactors? Volume?  Inoculation level?  Why did you use this yeast strains? Are there possible differences in different yeast cultures?

Tables 4 and 5. It is not clear the statistical analysis : Please indicate the statistically differences using different superscript letter. (P=0.05 LSD? Duncan? Fischer or other?)

line 207 Please specify the type of statistical analysis ( Analysis of variance??  and type of significant test)

Saccharomyces cerevisiae  for the first time and after using the abbreviation S. cerevisiae

Author Response

Responses to Reviewer 1# comments: The manuscript " Preparation of Selenium-enriched Yeast by re-using Discarded Saccharomyces cerevisiae in Beer Industry for Se-Supplemented Fodder Application " investigates on the selenium-enriched yeast coming from beer industry. The manuscript is well written and easy to follow. The results on the optimization of the biotechnological process are of interest in view of a possible use in fodder application. However there are some concerns that should be fixed: -Strain, Media and Growth Conditions section should be more detailed. What were the modalities of fermentation trials?? Flasks , Bioreactors? Volume? Inoculation level? Why did you use this yeast strains? Are there possible differences in different yeast cultures? Response: Many thanks for your precious comments. Modalities of fermentation trials: For laboratorial fermentation condition optimization, 50.0 mL of fermentation medium was put into 250 mL conical flask. For large-scale fermentation (data are not shown in the paper), 50.0 L of fermentation medium was put into 500 L fermentation tank. DB-yeast was obtained from Jinwei Beer Group co. LTD (Gangzhou, China) (approximately 80% moisture), and the optimized inoculation volume was 15.0%. The inaccurate description has been revised in “manuscript R1 2.1. Strain, Media and Growth Conditions” Indeed, some reaches also reported S. cerevisiae can be used for produce Se-enriched yeast (e.g. Yoshinaga, M. et al. Microorganisms 2018, 6: 81; Yin, H et al. LWT - Food Science and Technology 2010, 43: 666-669). However, in our study, we want to emphasize reuse of the discarded beer yeast (S. cerevisiae) in beer industry to produce Se-enriched yeast for Se-Supplemented fodder. -Tables 4 and 5. It is not clear the statistical analysis : Please indicate the statistically differences using different superscript letter. (P=0.05 LSD? Duncan? Fischer or other?) Response: Tables 4 and 5 have been revised. “a, Statistically significant for Na2SeO3 containing fodder compared with control (P value < 0.05); b, Statistically significant for Se-enriched DB-yeast containing fodder compared with control (P value < 0.05); c, Statistically significant for Se-enriched DB-yeast containing fodder compared with Na2SeO3 containing fodder (P value < 0.05)” The statistics significance was evaluated using Student’s t-test and P value < 0.05 was considered statistically significant. -line 207 Please specify the type of statistical analysis ( Analysis of variance?? and type of significant test) Response: The statistics significance was evaluated using Student’s t-test and P value < 0.05 was considered statistically significant. -Saccharomyces cerevisiae for the first time and after using the abbreviation S. cerevisiae Response: Yes, you are right, we have revised them in “manuscript R1”

Reviewer 2 Report

I encourage the authors to make the following revisions to improve the quality of the manuscript and to clarify some potential misleading phrases.

Title: The term “reusing discarded Saccharomyces cerevisiae in Beer industry” could be misleading. It gives the idea of using all these lees directly as fodder components. Is it the case? They are all taken into preservation media? I would suggest the authors to adapt this if this is not the case.

Abstract: Throughout the abstract and the entire manuscript, correct Na2SO3 when you refer to sodium selenite Na2SeO3.

Line 35: sulfhydryl instead of sulfydryl

Line 59: It would be better to describe to which region this cardiomyopathy is endemic to; whole country?

Line 107-109: These conditions were set after experimental results? Or after industrial fermentative conditions? How did you define these parameters?

Line 108: I guess you refer to degree Brix, I find it important to be mentioned or if it is another degree used.

Line 109: revise units, they are different from those in line 115. The selenium added was in form of salt, therefore the amount used was of selenium as sodium salt?

Line 155: what was the detection method followed? Would be interesting to know the analytical technique used together with the kit.

Line 158: same as previously mentioned.

Line 170: describe abbreviations SPF and NIH. Rephrase 168-171 to clarify idea.

Line 175: mice instead of mouse

Line 180: double word “nutrition”

Line 182: check units, kcal/kg calorie

Line 287: rephrase. It’s confusing.

Line 317: mice

Line 357: significantly

Line 375: organic instead of inorganic

Line 379: chosen

Line 379-381: rephrase, it is hard to understand as written

Lines 457-459: rephrase

Conclusions: by “fermentation accumulation” you mean?

Figure 4D: sulfhydryl

Results are interesting and support the fact that Selenium can be accumulated in yeast biomass after fermentation under experimental conditions. I nevertheless encourage the authors to improve English writing throughout the manuscript by having it corrected by a native speaker.

Author Response

Responses to Reviewer 2# comments: I encourage the authors to make the following revisions to improve the quality of the manuscript and to clarify some potential misleading phrases. -Title: The term “reusing discarded Saccharomyces cerevisiae in Beer industry” could be misleading. It gives the idea of using all these lees directly as fodder components. Is it the case? They are all taken into preservation media? I would suggest the authors to adapt this if this is not the case. Response: In our study, we want to emphasize reuse of the discarded beer yeast (S. cerevisiae) in beer industry to produce Se-enriched yeast for Se-Supplemented fodder. Title “reusing discarded Saccharomyces cerevisiae in Beer industry” revised into “re-using Discarded Saccharomyces cerevisiae from the Beer Industry for Se-Supplemented Fodder Applications”. In addition, we revised the inaccurate description of “2.1. Strain, Media and Growth Conditions”. “For laboratorial fermentation condition optimization, 50.0 mL of fermentation medium was placed into a 250 mL conical flask. The preliminary laboratorial fermentation conditions were 30 °C, pH 5.0, 9.0% degree Brix, 9.0% inoculation volume (4.5 mL of DB-yeast), 10 μg•mL-1 Se (22.35 μg/mL of Na2SeO3, purity 98.0%), and 9 h of Se adding time. The mixture was cultured at 160 rpm for 24 h.” For large-scale fermentation (data are not shown in the paper), 50.0 L of fermentation medium was put into 500 L fermentation tank. Fermentation conditions were 30 °C, pH 5.0, 9.0% of degree Brix, 15.0% of inoculation volume (7.5 L of DB-yeast), 30 μg•mL-1 of Se adding concentration (67.05 μg•mL-1 of Na2SeO3, purity 98.0%), 9 h of Se adding time, and 160 rpm cultured for 30 h. -Abstract: Throughout the abstract and the entire manuscript, correct Na2SO3 when you refer to sodium selenite Na2SeO3. Response: Yes, you are right. We have revised all of them in “manuscript R1”. -Line 35: sulfhydryl instead of sulfydryl Response: It has been revised in “manuscript R1”. -Line 59: It would be better to describe to which region this cardiomyopathy is endemic to; whole country? Response: such as Keshan disease (an endemic cardiomyopathy) and Kashin–Beck disease (a type of osteoarthritis) in China. -Line 107-109: These conditions were set after experimental results? Or after industrial fermentative conditions? How did you define these parameters? Response: We set these parameters according the experiences of yeast industrial fermentative condition. -Line 108: I guess you refer to degree Brix, I find it important to be mentioned or if it is another degree used. Response: Yes, you are right, we have revised it in “manuscript R1”. -Line 109: revise units, they are different from those in line 115. The selenium added was in form of salt, therefore the amount used was of selenium as sodium salt? Response: All units have been revised. In “manuscript R1”, we provided both Se and Na2SO3 amounts. -Line 155: what was the detection method followed? Would be interesting to know the analytical technique used together with the kit. Response: Sorry, some reagents of kit are not clear, in addition, consider of the article length, we didn't provided the detailed analytical technique of the kit in “manuscript R1”. -Line 158: same as previously mentioned. Sorry, some reagents of kit are not clear, in addition, consider of the article length, we didn't provided the detailed analytical technique of the kit in “manuscript R1”. -Line 170: describe abbreviations SPF and NIH. Response: SPF (Specific Pathogen Free) and NIH (National Institutes of Health) have been revised. -Rephrase 168-171 to clarify idea. Rewritten as follows: “According to the Horn method, Na2SeO3 fodder (4.46, 10.0, 21.5, and 46.4 mg/kg of Se content) and Se-enriched yeast fodder (2150, 4640, 10,000, 21,500 mg/kg of Se content) were evaluated using SPF (Specific Pathogen Free) NIH (National Institutes of Health) female and male mice (Guangdong medical Experimental Center, China).” -Line 175: mice instead of mouse Response: Revised. -Line 180: double word “nutrition” Response: Revised. -Line 182: check units, kcal/kg calorie Response: Deleted calorie. -Line 287: rephrase. It’s confusing. Rewritten as follows: “The pH of the Se-enriched DB-yeast extracted protein solution (from10.00 ± 0.06 decreased to 5.25 ± 0.07) decreased more slowly than that of the DB-yeast extracted protein solution (from 10.02 ± 0.04 decreased to 2.23 ± 0.07) following the addition of HCl (Figure 5A).” -Line 317: mice Response: Revised. -Line 357: significantly Response: Revised. -Line 375: organic instead of inorganic Response: Revised. -Line 379: chosen Response: Revised. -Line 379-381: rephrase, it is hard to understand as written Rewritten as follows: “DB-yeast was chosen as the Se accumulation carrier to avoid the potential risk of pollution and to limit wasting of resources.” -Lines 457-459: rephrase Rewritten as follows: “In the future, Se-enriched DB-yeast can also be applied as a fodder additive for pig, chicken, cattle, and etc. among others in an effort to produce more types of Se-enriched foods as a way to combat Se deficiency.” -Conclusions: by “fermentation accumulation” you mean? Response: We want to say inorganic Se (Na2SeO3) can be bio-transform to organic Se (Se-enriched DB-yeast) by re-using brewing industrial discarded S. cerevisiae accumulation after fermentation. Rewritten as follows: “To the best of our knowledge, the present study demonstrated, for the first time, inorganic Se (Na2SeO3) was bio-transformed to organic Se (Se-enriched DB-yeast) through fermentation accumulation by re-using brewing industrial discarded S. cerevisiae for Se-enriched fodder application.” -Figure 4D: sulfhydryl Response: It has been revised in “manuscript R1”. -Results are interesting and support the fact that Selenium can be accumulated in yeast biomass after fermentation under experimental conditions. I nevertheless encourage the authors to improve English writing throughout the manuscript by having it corrected by a native speaker. Response: Many thanks for your precious comments. English language of “manuscript R1” has been improved by Edanz Group.
